## Original Research Article

amplicon sequencing; CRISPR/Cas9; ICS1; salicylic acid.

**Author for correspondence:**
Detlef Weigel,
E-mail: weigel@weigelworld.org
Current address: Efthymia Symeonidi, Department of Biological Sciences, University of Utah, Salt Lake City, Utah 84112, USA; Julian Regalado, The Globe Institute, University of Copenhagen, Øster Voldgade 5-7, 1350 Copenhagen, Denmark.

# CRISPR-finder: A high throughput and cost-effective method to identify successfully edited *Arabidopsis thaliana* individuals

Efthymia Symeonidi, Julian Regalado, Rebecca Schwab, and Detlef Weigel

Department of Molecular Biology, Max Planck Institute for Developmental Biology, Tübingen, Germany

## Abstract

Genome editing with the CRISPR/Cas (clustered regularly interspaced short palindromic repeats/CRISPR associated protein) system allows mutagenesis of a targeted region of the genome using a Cas endonuclease and an artificial guide RNA. Both because of variable efficiency with which such mutations arise and because the repair process produces a spectrum of mutations, one needs to ascertain the genome sequence at the targeted locus for many individuals that have been subjected to mutagenesis. We provide a complete protocol for the generation of amplicons up until the identification of the exact mutations in the targeted region. CRISPR-finder can be used to process thousands of individuals in a single sequencing run. We successfully identified an *ISOCHORISMATE SYNTHASE 1* mutant line in which the production of salicylic acid was impaired compared to the wild type, as expected. These features establish CRISPR-finder as a high-throughput, cost-effective and efficient genotyping method of individuals whose genomes have been targeted using the CRISPR/Cas9 system.

## 1. Introduction

Genome editing has become a routine approach to investigate gene function *in vivo*. The recent development of CRISPR/Cas9-based systems has opened new doors for genome editing by simplifying the requirements for genome targeting, particularly in comparison to zinc finger nucleases and TALENs (Gaj et al., 2013). The system requires a nuclease (Cas9), an artificial single guide RNA (sgRNA), and a short sequence upstream of the sgRNA binding site called a Protospacer Adjacent Motif (PAM), which has the sequence 5′-NGG-3′ (Gasiunas et al., 2012; Jinek et al., 2012). Part of the sgRNA is complementary to 20 nucleotides in the targeted region of the genome, and the rest is responsible for the stabilization of the Cas9/sgRNA complex.

Interaction of the Cas9/sgRNA complex with the target site enables Cas9's endonuclease domains to generate a double-stranded break. Such breaks can be repaired through either the non-homologous end joining (NHEJ) or the homology-directed repair (HDR) pathway. NHEJ is error-prone, and can introduce small insertions or deletions that can lead to the disruption of open reading frames (Ma et al., 2004; Phillips & Morgan, 1994). In the case of HDR, a donor template complementary to the target needs to be present to introduce a specific region to the genome of interest (Gratz et al., 2014; Liang et al., 1998). The CRISPR/Cas9 and related systems have been used to generate knock-outs (Chang et al., 2013; Li et al., 2013), knock-ins (Auer et al., 2014; Platt et al., 2014) and to delete entire genes (Canver et al., 2014) in several species including the plant *Arabidopsis thaliana* (Feng et al., 2013; 2014; Hyun et al., 2015; Peterson et al., 2016).

While the generation of mutants using CRISPR/Cas9 is relatively easy, identification of desired mutations often requires screening many events. Two common approaches to screen for induced mutations are Sanger sequencing (Fauser et al., 2014; Feng et al., 2014) or the T7 Endonuclease1 (T7E1) assay (Ablain et al., 2015; Xie &Yang, 2013) applied to individual PCR products. Unfortunately, neither method provides immediately a precise identification of mutations in the desired region. For example, in the case of Sanger sequencing, the final readout merges the most abundant products in the template into one chromatogram (Sanger & Coulson, 1975; Strauss et al., 1986). This can lead to secondary peaks and sometimes a mixed signal due to other amplified molecules in the mixture, and can make it very hard to detect desired but rare events that might have occurred during editing. Confirmation of successful editing through subsequent cloning of a mixed PCR product followed by retrieval of bacterial colonies that carry the rare variant is time-consuming and expensive. Use of T7E1 can also yield inconclusive results due to its reliance on the T7 Endonuclease 1 to digest only fragments carrying mismatches (Mashal et al., 1995), which would miss homozygous mutants, as there are no mismatched fragments available for digestion. In addition, both techniques can be expensive for screening a large number of samples (>100).

These limitations led us to develop a robust and cost-efficient way of efficiently screening large numbers of samples. Here we introduce a high-throughput screening approach for identifying mutations using Illumina sequencing, called CRISPR-finder. We describe both the library preparation of the samples and the analysis pipeline for identifying editing events. The method is compatible with sequencing on different Illumina instruments (MiSeq and HiSeq300), and the adapter sequences could be modified for use on other platforms.

Our approach is inspired by an amplicon sequencing method previously developed for pooling samples for the analysis of microbiomes (Lundberg et al., 2013). In our approach, the amplicon libraries are generated through a two-step PCR amplification using specific combinations of oligonucleotides for the first step. During the PCRs, frameshifting nucleotides and one of 96 unique indices are added. Based on the unique combination of the frameshifting nucleotides and the barcode we were able to sequence hundreds of samples, for example >900 samples, in a single Illumina MiSeq run. To illustrate the accuracy and the precision of the method we describe how we identified and characterized a Cas9-free line with a mutation in the *ISOCHORISMATE SYNTHASE 1* (*ICS1*) gene.

## 2. Results

### 2.1. Target site identification

The aim of this study was to improve the speed of mutant identification with the CRISPR/Cas9 system. To demonstrate the efficacy of this new approach, the *ISOCHORISMATE SYNTHASE 1* (*ICS1*) gene was targeted in different *A. thaliana* accessions (Supplementary Table S1). *ICS1* encodes an enzyme involved in salicylic acid biosynthesis (Wildermuth et al., 2001).

The accessions of *A. thaliana* used in this study are from the first phase of the 1001 Genomes Project (Cao et al., 2011). The polymorph tool (http://polymorph.weigelworld.org) was used to align sequences of *ICS1* from the different accessions. Target sites without sequence variation among the accessions were identified to select the guide RNAs (Figure 1a).

Plants were transformed separately with the *ICS1* targeting construct (Supplementary Table S2). The primary transformants were found to have somatic editing events by using the CRISPR-finder genotyping pipeline. The selection of the transgene was based on glufosinate or the seed-specific expression of mCherry (Gao et al., 2016; Kroj et al., 2003).

### 2.2. Generation and sequencing of amplicons spanning CRISPR/Cas9 target sites

In order to quickly and unambiguously identify CRISPR/Cas9-induced mutations in a large number of plants, the targeted regions were amplified by PCR, attaching different barcodes for different individual plants, and then pools of barcoded PCR products were sequenced on an Illumina MiSeq (or HiSeq) instrument. The *ICS1* locus was targeted in different accessions to determine the efficacy of the method at different genetic backgrounds. Two sites were targeted in the gene, 72 bp apart for *ICS1*. The amplified regions were 211 bp long.

The amplicons were prepared based on a two-step PCR amplification protocol (Figure 1b–d). During the first round of amplification, the specific region of interest was amplified, and frameshifting nucleotides as well as part of the sequences required

for the fragment to hybridize to the flowcell for sequencing, the TruSeq adapters, were added. This was achieved by using specific combinations of oligonucleotides (Figure 1c) (Supplementary Table S3). The cleaned PCR product was used as a template for the second round of amplification, where the remainder of the TruSeq adapters and one of 96 barcodes were added (Lundberg et al., 2013) (Figure 1d) (Supplementary Table S3). Each PCR amplification step was carried out for 15 cycles.

The PCR products were quantified using the Quant-iT$^{TM}$ PicoGreen® dsDNA assay, normalized (described in Methods) and pooled (Supplementary Figure S1). For the sequencing on the MiSeq platform, the MiSeq reagent kit v2 (300-cycles or 500-cycles) (MS-102-2002 or MS-102-2003) was used (150 or 250 bp paired-end reads). The adapters were designed and chosen in order to be compatible with both MiSeq and HiSeq3000 platforms (Illumina, San Diego, USA); successful runs were carried out on both platforms.

### 2.3. Demultiplexing process

After sequencing, the pooled reads were demultiplexed in a two-step process. Ninety-six batches of combined samples were first identified via the indices that were located at the TruSeq adapters incorporated in the 2nd PCR amplification. This process was carried out with bcl2fastq (1.8.4) software, provided by Illumina, which also trims the sequence of the barcodes (https://my.illumina.com) (Figure 2a). The length of the reads downstream was 150 or 250 bp according to the kit that was used.

Subsequently, sequencing reads from different samples were mixed under the same barcode. In order to assign each read to the individual from which it came, we took advantage of the frameshifting nucleotides incorporated during the first step of the two-step PCR amplification. The first nine nucleotides from each read were used as 'secondary' barcodes to determine from which sample each read in the sequencing run originated; nine bases are sufficient to capture the unique frameshifting nucleotides used during the amplicon generation (Figure 2b). The specific combination of oligonucleotides during the first amplification generated a unique combination of forward-reverse oligonucleotide and barcode sequence information for each sample.

For binning of reads, the PlexSeq Python script (https://github.com/7PintsOfCherryGarcia/plexseq) was developed, which successfully demultiplexes >98% of reads in each dataset (Figure 2b). Since PlexSeq was run without allowing any mismatches of the 'secondary' barcodes, around 2% of the data could not be separated because of errors in PCR primers or errors introduced during the sequencing process; a loss of 2% of reads was deemed acceptable (Figure 2c–f). These unassigned reads are ignored in downstream analyses. A file with the expected 'secondary' barcodes needs to be provided in order for the script to successfully proceed with demultiplexing (Supplementary Figure S2). After PlexSeq, the sequences that were used as barcodes for demultiplexing were not trimmed, since they are part of the amplicons.

### 2.4. Analysis pipeline

After the demultiplexing process, each sample was genotyped in order to detect single nucleotide polymorphisms, as well as small insertions and deletions in the region of interest.

For each sample, reads were mapped back to the reference sequence for the gene of interest (Gene ID:843810) using the MEM

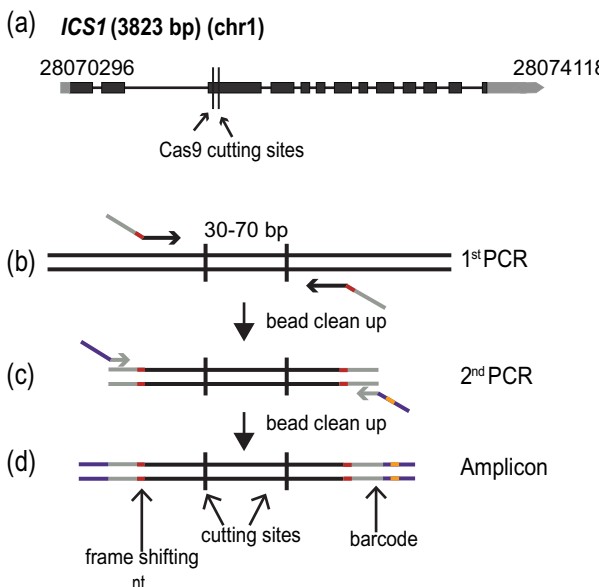

**Fig. 1.** Amplicon preparation. (a) Diagram of the targeted gene, *ISOCHORISMATE SYNTHASE 1* (*ICS1*). Black boxes indicate exons, and grey boxes untranslated regions. The arrow shows the direction of transcription. The numbers at the beginning and end of the gene correspond to the genomic coordinates. (b)–(d) Amplicon preparation. (b) The first PCR step to amplify a specific region of the genome. The oligonucleotide primers in this step fuse the first part of the TruSeq adapters (grey) and the frame shifting nucleotides (red). (c) The second PCR amplification adds the last part of the TruSeq adapters (purple) and one of the 96 barcodes (orange). (d) The final amplicon with frameshifting base pairs(s) (red), TruSeq adapters (grey and purple) and barcode(orange).

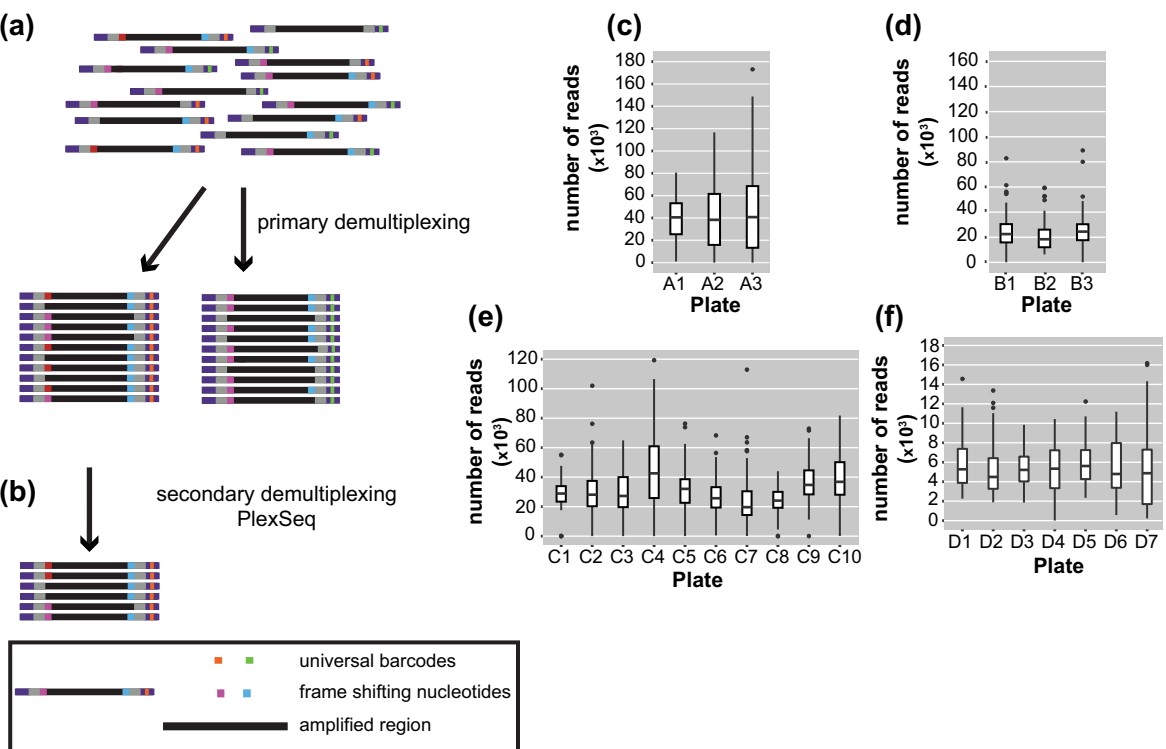

**Fig. 2.** Diagrams of the demultiplexing procedure and graphs representing the average number of reads/plate/run. (a) The primary demultiplexing step is carried out by the Illumina software and separates the samples based on the indices that are located within the adapter region into 96 pools. (b) The secondary demultiplexing script (PlexSeq) then assigns the reads to individual plants based on the frameshifting nucleotides. (c)–(f) Each graph shows the average number of reads per plate (≤96 samples) in each run. Each run consisted of different samples. For different runs, different numbers of plates were sequenced depending on the number of samples. (c) MiSeqrun010, (d) MiSeqrun024, (e) MiSeqrun046 and (f) MiSeqrun083. For exact number of samples per run see Supplementary Table S4.

algorithm of the BWA read mapping tool (Li et al., 2009) with standard parameters (Figure 3a). The resulting alignment files were genotyped with freebayes using standard parameters (Figure 3a)

(Garrison & Marth, 2012). The resulting VCF file was then filtered with vcftools (Danecek et al., 2011) to only keep samples in which high-quality variants were detected at regions of interest.

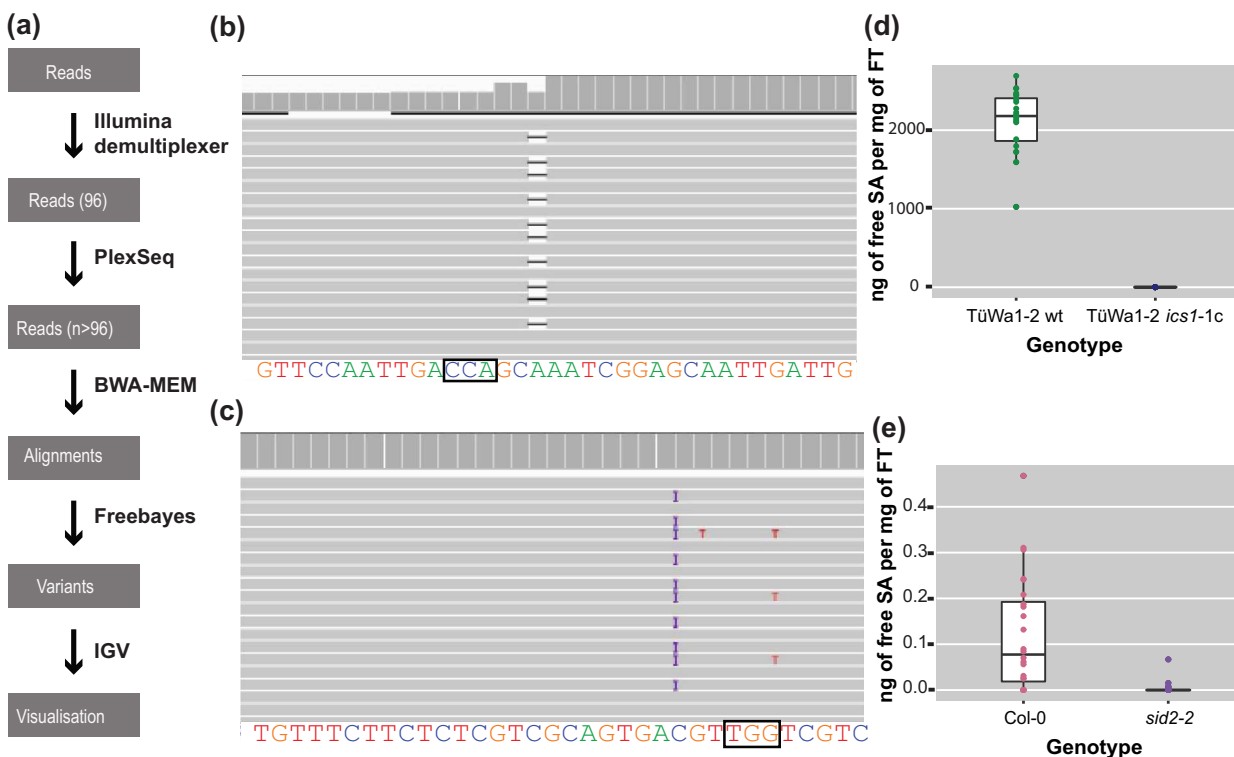

**Fig. 3.** The analysis pipeline and visualized alignments and SA levels of different genotypes. (a) Diagram of the analysis pipeline. BWA-MEM is used for the alignment and the Freebayes algorithm for variant calling. Finally, IGV is used for visualizing the alignments or the vcf files. (b) Alignment that shows a deletion visualized in IGV. On the top track in the coverage panel it is apparent how coverage is decreased at the location of the deletion. The black box indicates the location of the PAM site. (c) Alignment that shows a 1-bp insertion (purple 'I') in IGV. The black box indicates the location of the PAM site. (d) SA content of TüWa1-2 wild-type and the derivative TüWa1-2 *ics1*-1c mutant. (e) SA content of Col-0 reference wild type and isogenic *sid2*-2 mutant for comparison. *SID2* is a synonym for *ICS1*. Note the very different scale from (d). FT, Fresh Tissue. The whiskers of each boxplot indicate the spread of the data, and the line within the box corresponds to the median value of the data points. The measurements from plants grown in 23°C short-day conditions (8 h light/16 h dark) for 43 days.

At the end, integrative genomics viewer (IGV) (Robinson et al., 2011; Thorvaldsdóttir et al., 2013) was used for visual inspection of read mapping and variant calls (Figure 3b,c). All software was used with standard parameters unless otherwise noted. The required memory for the analysis can be 5–20 Gb depending on the output of the run.

### 2.5. Identifying mutations

Using CRISPR-finder, plants either heterozygous or homozygous for targeting events were identified. As a proof of concept for our approach, we targeted the *ICS1* gene in the TüWa1-2 background. The TüWa1-2 *ics1*-1c mutant was identified after screening more than 100 individuals. The parental genotype was originally collected in Germany and its phenotype shows extensive necrotic lesions on the leaves (Supplementary Figure S3), which can be attributed to extensive cell death. It was hypothesized that this is caused by elevated levels of SA. Using a biosensor assay, the SA content in plants was quantified (Defraia et al., 2008; Huang et al., 2006). As expected, the levels of free SA in the TüWa1-2 *ics1*-1c mutant were decreased in comparison to the ones in the wild-type parental lines (Figure 3d,e). These results demonstrate that our approach of screening can easily and rapidly identify individuals with targeted mutations that have the desired effect.

## 3. Discussion

We describe a high-throughput screening approach, called CRISPR-finder, that increases the accuracy and reduces the time and cost required for identifying CRISPR/Cas induced mutations (Figure 4). We generate barcoded amplicons of the targeted region through a two-step PCR amplification (Figure 1b–d). For each individual, a unique combination of frameshifting nucleotides and index sequence is used, which greatly increases the number of barcodes. An important consideration is that pooling of amplicons for sequencing can lead to unbalanced representation of samples. However, if we aim for average coverage of 1,000×, and assume that 10% of individuals provide 10× as many reads as aimed for, and 10% of individuals provide only one-tenth of the reads aimed for, a single MiSeq run (~15–20 million reads) would still provide sufficient coverage to analyze over 10,000 samples in a single run. Of course, the coverage can be adjusted to the needs of different experimental set ups.

For processing large numbers of samples, CRISPR-finder is a particularly cost-effective method. While with conventional assays such as Sanger sequencing and the T7E1 assay, costs scale linearly with the number of samples to screen, for CRISPR-finder in one sequencing pool, thousands of samples can be sequenced with high resolution and the cost per sample decreases as more samples are added to the pool. Additionally, 'spiking in' samples into another sequencing run to use only part of a flowcell's capacity is possible,

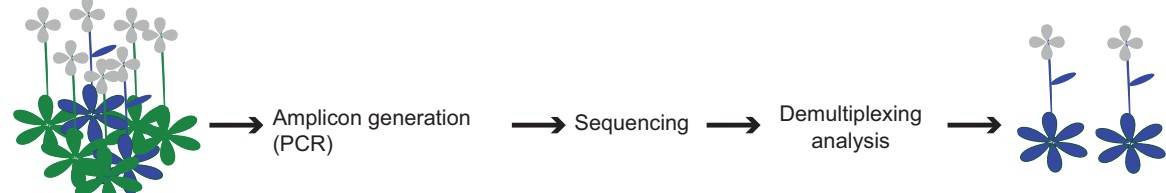

**Fig. 4.** Schematic representation of the screening pipeline. Starting from hundreds of samples the amplicon generation takes place by preparing the individuals for sequencing. By the end of the sequencing run the demultiplexing and analysis can take place that can lead to the identification of the desired edited individuals.

further increasing flexibility and reducing costs. We consider our method to be cost-efficient as soon as there are at least 100 individuals to be genotyped.

Finally, the first oligonucleotide set does not need HPLC purification, limiting the cost of the multiplexing procedure. This is because oligonucleotide primers are synthesized from 3′ to 5′, and truncations or errors will therefore be concentrated towards the ends of the amplicon during the first round of amplification. These ends serve as the binding sites for the oligonucleotides that are used for the second round of amplification, which will anneal despite minor errors and result in products with the correct adapter sequence.

While our method was developed for screening *A. thaliana* CRISPR/Cas9 mutagenized individuals, it can be easily adopted for any organism that has been genome edited using the CRISPR/Cas9 or related systems. The targeting of multiple sites can be accommodated with one amplicon, if the distance permits it (present study) or by generating one amplicon for each site. Note, however, that large deletions induced by CRISPR/Cas9 editing (Kosicki et al., 2018) would escape detection with our pipeline because the expected sequences for hybridization sites of the first set of oligonucleotides might not be present. In addition, while our method allows for efficient and precise genotyping and identification of individuals carrying the desired mutations, one still has to consider the downstream steps such as identification of transgene-free lines that faithfully inherit the mutation.

In conclusion, a full pipeline from DNA extraction to identification of individuals carrying mutations generated with the CRISPR/Cas9 system is described in detail – CRISPR-finder. Compared to more conventional methods (Sanger and T7E1 assay), large-scale amplicon sequencing is more robust and less expensive.

## 4. Material and methods

### 4.1. Plant growth

*A. thaliana* seeds were kept at −80°C overnight and then surface-sterilized with 70% ethanol and 0.05% (v/v) Triton X-100 for 5 min, followed by 100% ethanol for 5 min. Seeds were air-dried in a sterile hood until all residual ethanol had evaporated. Seeds were stratified in 0.1% (w/v) agar-agar for 7 days in the dark at 4°C prior to sowing on soil. Vernalization-requiring seedlings (highlighted with blue in Supplementary Table S1) were placed for seven weeks in 4°C short-day conditions (8 h light/16 h dark) and then transferred to 23°C long-day conditions (16 h light/8 h dark). For SA assays, plants were grown in 23°C short-day conditions (8 h light/16 h dark).

### 4.2. Plasmid generation

Constructs for plant transformation were generated using the GreenGate cloning system (Lampropoulos et al., 2013). The five different constructs used are described in Supplementary Table S2. The sgRNA constructs were generated as described in Wu et al. (2018), pEF016 (5′-AATCAATTGCTCCGATTTGC-3′) and pEF017 (5′-TTCTCTCGTCGCAGTGACGT-3′).

### 4.3. Plant transformation

Plants were transformed using the flora dip method as described by Clough and Bent (1998).

### 4.4. Selection of Cas9 transgene-free plants

Two selection markers were used, resistance to glufosinate ammonium (BASTA SL, Bayer Crop Science, Leverkusen, Germany) and AT2S3::mCherry (Gao et al., 2016). To select transgene-free plants that no longer carried BASTA resistance, leaves were brushed with a solution, diluted from the original stock (200 g/L) BASTA (1:1,000 or 1:2,000) (Bayer Crop Science, Leverkusen, Germany). The treatment caused leaves from plants without the transgene to become wrinkled and yellowish.

Seeds from plants that were carrying the AT2S3::mCherry (Kroj et al., 2003) cassette were screened for fluorescence or absence thereof under a LEICA MZFLIII Fluorescence stereoscope (Wetzlar, Germany) with a SOLA 365 SM Light Engine© lamp (Lumencot, Beaverton, OR).

We consider as good practice that one confirms the absence of the transgene with a genotyping approach for any line that will be used for subsequent experiments.

### 4.5. DNA isolation

Genomic DNA was extracted following a published protocol (Edwards et al., 1991), with an additional ethanol wash. DNA was resuspended in 100 μl of ddH$_2$O.

### 4.6. Salicylic acid quantification

The protocol was adapted from Marek et al. (2010). Fresh tissue was collected and frozen at −80°C overnight. For every 175 mg of fresh tissue, 250 μl of 0.1 M pH 5.5 sodium acetate was added post grinding for further vortexing. *Acinetobacter* sp. ADPWH_lux strain was used (Huang et al., 2006) for the quantification of salicylic acid. Overnight culture of *Acinetobacter* sp. ADPWH_lux at 37°C was diluted (1:20) and grown at 37°C while shaking at 200 rpm until it reached OD$_{600}$ of 0.4. For measuring free and 2-O-$\beta$-D-glucoside (SAG) SA, plant crude extract from the samples was incubated at 37$^o$C for 1.5 h with 0.4 U/μl of $\beta$-glucosidase prior to measurement.

Black Optiplates (96 wells, ref:655906; Greiner Bio-One, Kremsmünster, Austria) were used for the measurements. They were loaded with 50 μl of LB, 60 μl of the cell culture and 30 μl of the plant extract. Standards were prepared with 50 μl of LB, 60 μl of the cell culture, 10 μl of known SA concentrations and 20 μl of plant extract from *35S::NahG* plants as control (Col-

0 background) (prepared the same way as the samples). The plates were incubated at 37°C for 2 h without shaking and the luminescence was measured using the Iinfinite F200 instrument (TECAN, Männedorf, Switzerland) and the i-control 1.12 software.

### 4.7. Amplicon library preparation

The amplicon libraries were generated with a two-step PCR protocol. The first reaction consisted of 1 μl of genomic DNA as template, 0.5 μM forward oligonucleotide (G-40604/G-40605/G-40606/G-40606/G-42015), 0.5 μM reverse oligonucleotide (G-40607/G-40608/G-40609/G-42016), 1× Phusion HF buffer (1.5 mM MgCl₂) (Thermo Fisher Scientific, Waltham, MA), 0.2 mM dNTPs (Thermo Fisher Scientific, #R0182, Waltham, MA) and 0.02 U/μl Phusion High-Fidelity DNA polymerase (Thermo Fisher scientific, #F530, Waltham, MA) to a final volume of 25 μl.

The second PCR amplification consisted of 2.5 μl of the cleaned PCR product of the previous reaction, 0.5 μM forward oligonucleotide (G-40610), 0.25 μM reverse oligonucleotide that had one of the 96 indices (Lundberg et al., 2013), 1× Phusion HF buffer (1.5 mM MgCl₂) (Thermo Fisher Scientific, Waltham, MA), 0.2 mM dNTPs (Thermo Fisher Scientific, #R0182, Waltham, MA) and 0.02 U/μl Phusion High-Fidelity DNA polymerase (Thermo Fisher Scientific, #F530, Waltham, MA) to a final volume of 25 μl.

Sequencing libraries were prepared using Q5® High-Fidelity DNA polymerase (New England BioLabs, #M0491, Ipswich, MA) in a final concentration of 0.02 U/μl along with 1× Q5 reaction buffer (2 mM MgCl₂). The rest of the reaction components (DNA template, dNTPs) remained the same.

The MJ Research PTC225 Peltier (Marshall Scientific, Hampton, NH) or the BIO-RAD C1000 Touch (Hercules, CA) thermal cyclers were used. The PCR programs had 15 cycles in which the denaturing temperature was 94°C for 30 s, followed by annealing at 60°C for 30 s, and extension at 72°C for 10 s for program 1, and 15 s for program 2. A final extension step was at 72°C for 2 min.

### 4.8. Bead clean up

For the generation of the amplicon libraries, two bead-based clean-up steps were carried out using SPRI beads (Magnetic Speed-Beads™, GE Healthcare No.:65152105050250, Chicago, IL). The first PCR product was cleaned using a ratio of 1:0.9 (reaction:beads v/v) and resuspended in 17 μl of ddH₂O. The second PCR product was cleaned using the same ratio of beads and resuspended in 27 μl. The ratios of clean ups were chosen after optimization.

### 4.9. Quant-iT™ PicoGreen® dsDNA assay

Amplicons were quantified using the Quant-iT™ PicoGreen (Invitrogen, Carlsbad, CA) dsDNA assay. One microlitre of each amplicon was used according to the manufacturer's instructions for the quantification. The samples were prepared in black 96 well, F-bottom, non-binding microplates (96 wells, ref:655906; Greiner Bio-one), and the TECAN Infinite M200 PRO plate reader was used for all the measurements using the Magellan 7.2 software.

### 4.10. Pooling (Supplementary Figure S1)

To roughly normalize samples when pooling, the DNA concentration of all samples in each 96 well plate was first measured fluorometrically (PicoGreen assay). First, all the 96 samples from each plate were pooled, creating subpools. From samples with concen-

trations less than half of the mean, 6 μl were taken. From samples with concentrations more than twice the mean, 1.5 μl was taken. For all other samples falling between these extremes, 3 μl was taken. After each plate was pooled in this way, the subpools representing entire plates were again measured fluorometrically (Qubit dsDNA-HS assay) (Thermo Fisher Scientific, Waltham, MA) and pooled in an equimolar manner to create a final pool containing all samples. The concentration of the subpools and the final pool were evaluated using the Qubit dsDNA-HS assay. Each pool was analyzed on the Agilent 2100 Bioanalyzer (Agilent Technologies, Santa Clara, CA) according to the manufacturer's instructions. DNA1000 chips were used for the amplicon libraries.

### 4.11. Illumina MiSeq sequencing

The libraries were diluted for Illumina sequencing following manufacturers' protocols and sequenced on the MiSeq platform using MiSeq reagent kit v2 (300-cycles) (MS-102-2002) for the MiSeq 010, 024 and 046 and MiSeq reagent kit v2 (500-cycles) (MS-102-2003) for the MiSeq 083.

## Acknowledgements

We thank Derek Lundberg for discussions, suggestions regarding PCR and library preparation and comments on the manuscript, Talia Karasov for comments on the manuscript, Christa Lanz and Julia Hildebrandt for technical help sequencing, Ilja Berzukov for Illumina demultiplexing.

**Financial Support.** Funding was provided by the Bayer Science and Education Foundation (to ES) and the Max Planck Society.

**Conflict of Interest.** The authors declare no competing interests.

**Authorship Contributions.** E.S., R.S. and D.W. planned the study. R.S. and E.S. prepared initial plasmids for the CRISPR/Cas9 system. E.S. performed all cloning, plant transformations, DNA extractions and prepared all the libraries. JR designed and wrote PlexSeq for sample demultiplexing. E.S. carried out data analysis. E.S. conducted *Hpa* in infections. E.S. wrote the manuscript. E.S., J.R., R.S. and D.W. revised the manuscript.

**Data availability Statement.** All data in this manuscript has been deposited in the European Nucleotide Archive. It can be accessed under the project number PRJEB39078.

**Supplementary Materials.** To view supplementary material for this article, please visit http://dx.doi.org/10.1017/qpb.2020.6.

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
