## [Reviewer Report]

Dear Olivier,

We would like to submit the attached manuscript “CRISPR-finder: A high throughput and cost effective method for identifying successfully edited A. thaliana individuals” for publication in Quantitative Plant Biology. 

As you know, CRISPR/Cas9 technology has become the most successful genome editing tools during the last years. Depending on the system, accurate identification of edited individuals is one of the bottlenecks in applying this method in practice, because a large number of individuals needs to be genotyped. Screening using Sanger sequencing or T7 Endonuclease I (T7E1) mismatch assays is expensive and lacks resolution, often adding significant cost and/or experimental burden.

We introduce a high throughput and cost-effective approach for identifying the exact mutations in edited individuals, based on a two-step PCR amplification of the targeted region. We demonstrate the precision and versatility of the method by using it in different A. thaliana accessions. We anticipate that the technique will be widely adopted because of its accuracy and its simplicity. A preprint of this study has been uploaded on bioRxiv (https://www.biorxiv.org/content/10.1101/2020.06.25.171538), and quite a few colleagues have already started to implement the method we describe.

We declare no competing interests. Knowledgeable reviewers for this work include Luca Pinello (Massachusetts General Hospital), Holger Puchta (Karlsruhe Institute of Technology), Astrid Cruaud (INRA), MItchell L Sogin (The Marine Biological Laboratory) and Frederic Bushman (U Penn). We are looking forward to hearing from you.

Yours truly,

Detlef Weigel

Executive Director, Max Planck Institute

---

## [Reviewer Report]

*Comments to Author*: The manuscript describes an experimental and analysis pipeline for being able to screen large numbers of plants for desired gene editing events. The work addresses a very important issue, as it is currently much easier to induce gene edits using CRISPR/Cas9 than to screen for desired gene editing events in the resulting progeny. The authors also present a nice (and impressive) proof of concept for the potential of efficient gene editing. A well worked out pipeline such as described here is timely and will certainly be of high interests for many plant scientists. Moreover, the manuscript is written and illustrated very well and easy to follow.

Despite my enthusiasm for the manuscript, I am not sure if the manuscript is a good fit for Quantitative Plant Biology. In its present form it is a method paper, but the scope of the journal is described as publishing “ground-breaking discoveries and predictions in quantitative plant science”. One possibility to improve with regards to the journal match might be to actually use the data that should exist according to the manuscript: The same gene edits were performed in 48 different accessions and using two different Cas9 versions. Perhaps, we can learn something about the impact of the genetic background and/or these different Cas9 versions on gene edits. Moreover, I am sure scoring the phenotypes of all these mutants in different accessions would reveal exciting things. However, it is of course it is very likely that these findings will be part of another study.

In addition to this more significant issue, there are a couple of minor issues:

1. In the current form of the manuscript, it is somehow unclear why the different accessions/Cas9 versions are presented in the manuscript. Results for these accessions or Cas9 versions are not presented at all (maybe it is included in the database deposition, but I couldn’t access it).

2. It would be nice to have a bit more discussion on the limits of the approach: Would it work with multi-gRNA promoter editing (it seems that larger INDELs etc. can be frequently caused by that), what bottlenecks remain once the genotyping problem is solved?

3. It might be helpful to explain some of the abbreviations or jargon terms. For instance, L93 MiSeq is not explained, or Trueseq adapters are not.

4. Fig 1a: Are the numbers at the gene model the genomic coordinates?

5. Fig 2c-f: The sample names are the same; Are they repeats or different samples? Also, the y-axis should have scientific notation (20k might confuse some readers)

6. L189: Typo: “Figure”

7. Fig 3d,e: Spell out or define FT in y-axis label; Define properties of the box plot (what are the whiskers, box, vertical line indicating)

8. L233: “free SA in the TüWa1-2 c-ics1-1 mutant were significantly lower” specify stat. test and test result either in Figure 3 or in text;

9. L307ff ”Selection of Cas9-free plants”: How can one be sure that herbicide resistance gene or FP are not silenced (thus the plants would not be Cas9 free)?

10. L347: Typo “Thermo FIsher"

11. L375,L377: Unclear what the ratio refers to.

---

## [Reviewer Report]

*Comments to Author*: Review Symeonidi et al., september 2020

This is a technical paper providing a method to accurately detect small mutations triggered by the CRISPR-Cas9 tool in Arabidopsis using a cost-efficient NGS method. The authors describe a 2-steps PCR-based multiplexing method in which they amplify the region of interest with a first set of unique frameshifting barcodes. Then after pooling, a second PCR with a set of Illumina barcodes is performed. The information is then deconvoluted in two steps to assign reads to individual plants.

The authors describe the bioinformatics method they set up to separate the reads in two steps. One step with the standard Illumina demultiplexing tool, and then their homemade method (Plexseq) which looks up the secondary (frameshift) barcode as an exact match in the 5’ end of the first PE read, and then, if applicable (PE data) looks at the exact match et the 5’ end of the secondary read.

For the moment, the bioinformatics tool does not allow errors on the secondary read 5’ end, which results in the loss of a few % of the reads. However, this minor amount of “lost “ reads does not affect the final characterization of the DNA sequence in transformed plants.

This is an interesting paper which may help plant groups save time to characterize their different CRISPR lines. As most groups are massively using this emerging tool, they realize that genotyping by Sanger becomes really costly (both in time and money). Hight thoughput methods such as the one decribed in here represent interesting alternatives. Double amplicfications and deconvolution have already been described for other species (Brocal et al., . BMC Genomics. 2016;17:259, for example), but this time the authors cleverly reused/recycled frameshifting barcodes, which were previously described in environmental sequencing for a totally different purpose. I liked this agile approach.

I thus recommend publication for this work, however I have one major comment on how the “wet” steps are described:

As it is presented, it is not easy to clearly understand the “how” of the two steps PCR protocol. This part should be more clearly explained so that the reader can understand the interest of the method. My misunderstanding came from the reading of the original paper on frameshifting nucleotides (Lundberg et al., 2013). In this study, frameshifting nucleotides have been used to capture molecular diversity in complex environmental DNA mixes to avoid biases generated during PCR amplification from complex mixtures. The use of frameshifting primers in this study has a totally different goal finally. I initially thought that these nucleotides were used to capture molecular diversity from each individual with contains a range of somatic mutations, and got confused. Then by looking at the suppl tables describing the 8 different frameshifting nucleotides I understood that these 8 different primers were used to amplify DNA from 8 different individuals. Then these 8 samples are pooled in 1 single batch for the second PCR round. Up to 96 batches of 8 individuals can then be sequenced in one run. Am I right? So maybe this part should be more carefully rephrased so as to make it more explicit and clear.

Are the sample tagged in unique combination (i.e. sample 1 is amplified with G-40604 and G-40607, allowing the same frameshift barcode on both ends) or in combinatorial multiplexing?

Overall, no description of the pooling design is provided, and no description of the actual number of individuals which could be genotyped in the 4 different runs is provided. I think figure 1 should be redesigned to incorporate a description of the exact pooling strategy so as to view the actual scale of the study.

I have a few more minor comments:

There is little information on whether single end reads or paired end are provided by the MiSeq (only one mention of the use of the MiSeq reagent kit V2 which comes rather late in the manuscript and remains elusive on the type of output that was chosen : SE or PE, read length?). What was the average “useful” read length after trimming the to consecutive barcodes? Which % of the reads actually mapped on the amplicon sequence? The authors should be more precise.

Figure 2 should be redesigned with colors matching figure 1 (first PCR with red and grey tags, then second PCR specific colors).

Finally, there are not enough details on sample numbers, sequencing results reads and finally with so little precision the reader cannot really estimate whether this strategy can be cost effective or not, and what is the approximate number of samples from which it becomes really interesting to use this NGS method.

---

## [Reviewer Report]

*Comments to Author*: There were extenuating circumstances that led to a delay in the completion of this review cycle; I appreciate your patience. The result of this evaluation are overall positive, with reviewer 1 requesting minor revisions and reviewer 2 requesting revisions that should not require additional experimentation or analysis. The suggested revisions will make this work more accessible to a broader and interdisciplinary audience. I’d advise to make the required corrections, and add more context (e.g. in the discussion) on how how your protocol will be useful for quantitative plant biology papers (following the reviewer comments). It is my intention to evaluate your textual revisions myself before a final decision.

---

## [Reviewer Report]

Dear Olivier,

Many thanks for having our manuscript “CRISPR-finder: A high throughput and cost effective method for identifying successfully edited A. thaliana individuals” considered for publication in Quantitative Plant Biology. 

We were very pleased with the positive comments and have hopefully addressed all of them satisfactorily. We are looking forward to hearing from you.

Yours truly,

Detlef Weigel

Executive Director, Max Planck Institute

---

## [Reviewer Report]

*Comments to Author*: I commend the authors on this revised manuscript.

The authors have addressed all points that I had raised, with one minor exception that I suggest they revise:

While they have streamlined the manuscript and removed information about the approaches (accession, Cas9 versions) for which no data is presented, in line 234 (methods section) they still refer to two Cas9 versions. It is not clear which one is the Cas9 version for which the results are reported. Please clarify.

---

## [Reviewer Report]

*Comments to Author*: All the issues I pointed out in the first review have been adressed,all added figures and table improve the manuscript, which is clear and easily understandable .

---

## [Reviewer Report]

*Comments to Author*: Thanks for the revised version. Please make sure you address the minor comment of Reviewer 1.